# CLAIR: Evaluating Image Captions with Large Language Models

**David M. Chan, Suzanne Petryk, Joseph E. Gonzalez, Trevor Darrell, John Canny**

University of California, Berkeley

{davidchan,spetryk,jegonzal,trevordarrell,canny}@berkeley.edu

## Abstract

The evaluation of machine-generated image captions poses an interesting yet persistent challenge. Effective evaluation measures must consider numerous dimensions of similarity, including semantic relevance, visual structure, object interactions, caption diversity, and specificity. Existing highly-engineered measures attempt to capture specific aspects, but fall short in providing a holistic score that aligns closely with human judgments. Here, we propose CLAIR[1], a novel method that leverages the zero-shot language modeling capabilities of large language models (LLMs) to evaluate candidate captions. In our evaluations, CLAIR demonstrates a stronger correlation with human judgments of caption quality compared to existing measures. Notably, on Flickr8K-Expert, CLAIR achieves relative correlation improvements over SPICE of 39.6% and over image-augmented methods such as RefCLIP-S of 18.3%. Moreover, CLAIR provides noisily interpretable results by allowing the language model to identify the underlying reasoning behind its assigned score. Code is available at https://davidmchan.github.io/clair/.

## 1 Introduction & Background

Automatically evaluating the quality of image captions is challenging. There are many dimensions to consider, such as grammatical quality, semantic relevance, correctness, and specificity, among others. To ensure fair evaluations, most image captioning works employ a suite of measures, each capturing different aspects. For instance, n-gram-based measures like BLEU (Papineni et al., 2002) or CIDEr (Vedantam et al., 2015) broadly measure content overlap, SPICE (Anderson et al., 2016) compares scene graph structures, and CLIPScore, TIFA, SeeTrue and VPEval (Hessel

---

[1]Meaning "clear" in French, in line with other colorful metric names (Papineni et al., 2002; Lin, 2004; Lita et al., 2005).

Figure 1: CLAIR: a (surprisingly simple) large language model-based measure for image caption evaluation. We find that CLAIR not only correlates strongly with human judgments of caption quality but can also generate interpretable reasons for the generated scores.

et al., 2021; Hu et al., 2023; Yarom et al., 2023; Cho et al., 2023) directly incorporate visual information. Unfortunately, while significant strides have been made in automated evaluation, human preference studies remain the most reliable (yet costly) source of caption evaluation.

Fortunately, recent advances in large language models (LLMs) have opened new avenues for automatic evaluation. Models trained with reinforcement learning from human feedback (RLHF, Christiano et al. (2017)) or similar methods are particularly useful for open-ended evaluation tasks, including image captioning, due to their explicit training to align with human preferences.

In our work, paralleling several recent works which find that LLMs can act as effective "judges" for selecting the better answer from two candidates (Bubeck et al., 2023; Dettmers et al., 2023; Chiang et al., 2023), we explore the ability of LLMs to evaluate caption quality in the multimodal setting.

We introduce CLAIR (Criterion using LAnguage models for Image caption Rating), a measure which scores a candidate caption based on the likelihood that it describes the same image as a set of references by directly asking an LLM to produce a numeric rating. We further query the LLM to provide a *reason* for its score, providing a level of interpretability to the scalar rating. As far as we are aware, this is the first paper to explore replacing measures of *semantic text quality* with directly obtained LLM judgments, however concurrently, Zheng et al. (2023) have shown that directly providing an answer rating can align highly with human preferences on a range of standard language-based tasks, such as conversational instruction following.

Through several experiments on captioning datasets such as MS-COCO (Xu et al., 2016), Flickr8k (Mao et al., 2014), and PASCAL-50S (Vedantam et al., 2015), we find that CLAIR correlates surprisingly well with human preferences, outperforming prior captioning measures. We additionally propose CLAIR$_E$, where we Ensemble the outputs of several LLMs by taking the average score, leading to further improvements.

Despite a simple pipeline using an LLM prompt with minimal output parsing, CLAIR's strong correlation with human preferences suggests that it captures multiple dimensions of caption similarity at once – a feature that prior measures struggle to achieve alone. More generally, CLAIR demonstrates how language-only models can evaluate vision-language tasks. We show LLMs can provide not only reliable scalar ratings but also corresponding reasoning for a given rating, offering a valuable combination of accuracy and interpretability.

## 2 CLAIR: LLMs for Caption Evaluation

In CLAIR, we adapt the zero-shot in-context learning approach described in Brown et al. (2020) to score candidate captions with large language models (LLMs). This involves converting the caption evaluation problem into a human-readable text completion task which is solved by the LLM. Using the prompt in Figure 1, CLAIR first generates completions from the LLM and then parses those completions into both candidate scores and an explainable reason for the score. We use a greedy sampling method ($t = 0$) to encourage reproducibility in the results, while acknowledging the inherent nondeterminism in LLMs (see section 4). CLAIR's

experimental implementation is surprisingly simple: it uses no in-context examples (is entirely zero-shot), and default inference parameters for the APIs. See Appendix B for further implementation details.

The choice of language model directly affects the quality of the CLAIR measure – more accurate models should produce evaluations that align better with human judgment. We explore three language models: GPT-3.5 (ChatGPT) (OpenAI, 2022), Claude (Instant) (Bai et al., 2022), and PaLM (Chowdhery et al., 2022). Unfortunately, we found for several open-weight language models including Koala (Geng et al., 2023) and Vicuna (Chiang et al., 2023) that CLAIR aligned poorly with human judgment.

As CLAIR is language model-agnostic, we can leverage the different distributions learned by each model and combine their decisions in an ensemble approach we term CLAIR$_E$. We calculate individual CLAIR scores for each model and compute an unweighted average to obtain the ensemble score.

**Benchmark measures:** We benchmark against several existing measure of caption similarity. BLEU (Papineni et al., 2002), ROUGE (Lin, 2004), METEOR (Agarwal and Lavie, 2008) and CIDEr (Vedantam et al., 2015) all primarily measure n-gram overlap (however have different weighting schemes between n-grams, and across precision/recall). We also compare against SPICE (Anderson et al., 2016), which compares caption parse trees and focuses on matching perceived action and object relationships in addition to n-grams. While the aforementioned measures are commonly reported in image captioning works, we also compare against several more modern measures, including BERT-Score (Zhang et al., 2020) (which measures the distance between BERT embeddings in the text), BERT-Score++ (Yi et al., 2020) (which fine-tunes BERT for image captioning), LEIC (Cui et al., 2018) and NUBIA (Kane et al., 2020) (which are custom trained models for image caption evaluation), TIGEr (Jiang et al., 2019) (which is a model trained for caption evaluation which takes into account the original image context), and CLIP-Score (Hessel et al., 2021) which uses the recent CLIP (Radford et al., 2021) model for reference-free evaluation.

## 3 Evaluation & Discussion

To evaluate the quality of the measure, we run several evaluations that compare scores generated

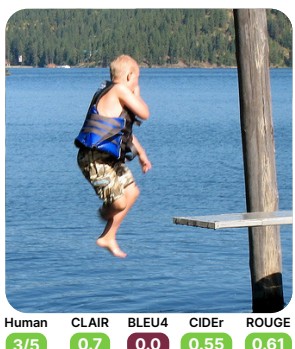 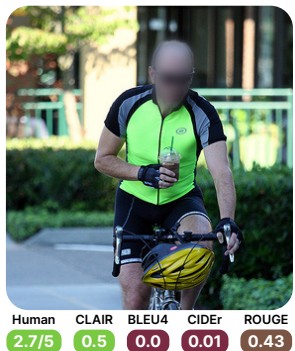 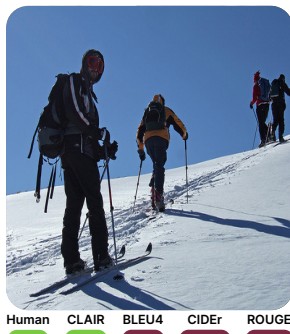

| | Human | CLAIR | BLEU4 | CIDEr | ROUGE |
|---|---|---|---|---|---|
| | 3/5 | 0.7 | 0.0 | 0.55 | 0.61 |

**Candidate**
A boy jumps into the blue pool water.

**CLAIR Reason**
The candidate set and reference set both mention a boy jumping into water, but the candidate set does not provide as much detail about the boy's appearance or the location of the pool. Additionally, the reference set includes captions about diving off a pier and holding one's nose, which are not mentioned in the candidate set.

**References**
"A blonde boy wearing a blue life vest and camo shorts jumps off a diveboard into a blue lake."
"A boy diving backward off a pier in a large lake."
"A boy in a blue life jacket jumps into the water."
"A boy in a blue life jacket jumps off a board into the lake."
"A boy is holding his nose and jumping off a diving board backwards into a lake."

| | Human | CLAIR | BLEU4 | CIDEr | ROUGE |
|---|---|---|---|---|---|
| | 2.7/5 | 0.5 | 0.0 | 0.01 | 0.43 |

**Candidate**
A cyclist in a helmet and yellow jersey.

**CLAIR Reason**
The candidate set only mentions a cyclist in a helmet and yellow jersey, while the reference set describes a man in a green shirt carrying a drink while riding a bicycle. Although there are some similarities, such as the fact that both sets mention a person riding a bike, the differences in the details make it uncertain whether they are describing the same image or not.

**References**
"A biker enjoys a coffee."
"A man in a bright green shirt and sunglasses is riding a bicycle through the streets whilst drinking a latte."
"A man in a bright green shirt riding a bicycle in a paved courtyard , carying a drink."
"A man rides on his bike with one hand and holds a drink with the other."
"A man riding a bike wearing a green shirt with a drink in his hand ."

| | Human | CLAIR | BLEU4 | CIDEr | ROUGE |
|---|---|---|---|---|---|
| | 2/5 | 0.4 | 0.0 | 0.00 | 0.14 |

**Candidate**
Person snowboarding at a ski slope.

**CLAIR Reason**
The candidate caption mentions only one person snowboarding, while all the reference captions mention multiple people skiing or climbing a mountain. Additionally, the candidate caption does not mention any details about the mountain or the surroundings, which are present in the reference captions.

**References**
"a group of skiers going up a mountain."
"Four cross-country skiers climb uphill."
"Four people climbing a hill in the snow."
"Four skiers walking up a snow covered hill."
"four skiers climbing snow drapped mountain."

Figure 2: Several qualitative examples of CLAIR from the Flickr8K-Expert dataset. CLAIR not only correlates better with human judgments of caption quality but also provides detailed explanations for its score. CLAIR scores normalized by 100.

by CLAIR to both human judgments of caption quality and other image captioning evaluation measures. We additionally provide several qualitative examples in Figure 2. A unique benefit of CLAIR is that it provides not only numeric scores but is also introspectable, as it can identify which details in the candidate caption set match the reference set.

**Sample-level human correlation:** We first ask the question, how well does CLAIR correlate with human judgments of caption quality at a sample level? We do so by exploring the performance on three datasets, COMPOSITE, Flickr8K-Expert, and MS-COCO (See Appendix B for details).

The results of our sample-level correlation experiments are shown in Table 1. We can see that CLAIR outperforms language-only measures (e.g., 0.604 to 0.449 for BERT-S++), and in most cases, outperforms vision-augmented measures. $CLAIR_E$ achieves strong sample-level correlation on all datasets; for instance, $CLAIR_E$ closes the gap to inter-human agreement by 0.097 over vision-based measures and 0.132 over language-based measures. The improvements of $CLAIR_E$ over CLAIR suggest that each language model may have some bias (similar to each human), yet the ensemble of models correlates more strongly with human judgments. A reasonable concern might be that the models underlying existing approaches are significantly smaller than those in CLAIR, and trained on less

Table 1: Sample-level correlation (Kendall's $\tau$) with human judgments. All p-values $< 0.001$. *: Model has access to additional visual context. Results for LEIC, BERT-S++, TIGEr, and NUBIA are drawn from their original work.

| | Dataset | | |
|---|---|---|---|
| Measure | COMPOSITE | Flickr8K | MS-COCO |
| BLEU@1 | 0.313 | 0.323 | 0.265 |
| BLEU@4 | 0.306 | 0.308 | 0.215 |
| ROUGE-L | 0.324 | 0.323 | 0.221 |
| BERT-S | 0.301 | 0.392 | 0.163 |
| METEOR | 0.389 | 0.418 | 0.239 |
| CIDEr | 0.377 | 0.439 | 0.262 |
| SPICE | 0.403 | 0.449 | 0.257 |
| BERT-S++ | 0.449 | 0.467 | - |
| NUBIA | - | 0.495 | - |
| LEIC* | - | 0.466 | - |
| TIGEr* | 0.454 | 0.493 | - |
| CLIP-S* | 0.538 | 0.512 | 0.217 |
| RefCLIP-S* | 0.554 | 0.530 | 0.305 |
| RefCLIP-X* | 0.523 | 0.549 | 0.274 |
| CLAIR | | | |
|   + GPT3.5 | **0.604** | 0.616 | 0.296 |
|   + Claude | 0.542 | 0.563 | 0.320 |
|   + PaLM | 0.580 | 0.546 | 0.355 |
| $CLAIR_E$ | 0.592 | **0.627** | **0.374** |
| Inter-Human | - | 0.736 | - |

data. To address this, we introduce and compare against RefCLIP-X, which replaces the CLIP model in RefCLIP with a CLIP ViT-bigG/14 model trained on LAION 2B (Ilharco et al., 2021). Even in this case, CLAIR demonstrates significantly improved performance.

Table 2: System-level correlation between the average CLAIR score and human model evaluation for 5 models trained and evaluated on MS-COCO. All p-values < 0.05.

| Measure | Kendall's $\tau$ | Spearman's $\rho$ | Pearson r |
|---|---|---|---|
| BLEU@1 | 0.399 | 0.600 | 0.706 |
| BLEU@4 | 0.799 | 0.899 | 0.910 |
| ROUGE-L | 0.600 | 0.700 | 0.792 |
| METEOR | 0.600 | 0.700 | 0.666 |
| CIDEr | 0.399 | 0.600 | 0.856 |
| SPICE | 0.399 | 0.600 | 0.690 |
| CLAIR | | | |
| + GPT3.5 | 0.799 | 0.899 | 0.869 |
| + Claude | **1.000** | **1.000** | 0.868 |
| + PaLM | **1.000** | **1.000** | **0.954** |
| CLAIR$_E$ | **1.000** | **1.000** | 0.903 |

Table 3: Accuracy of measures when matching human decisions for PASCAL-50S (5 reference captions). *: Model has access to additional visual context.

| Measure | HC | HI | HM | MM | All |
|---|---|---|---|---|---|
| BLEU@1 | 51.20 | 95.70 | 91.20 | 58.20 | 74.08 |
| BLEU@4 | 53.00 | 92.40 | 86.70 | 59.40 | 72.88 |
| ROUGE-L | 51.50 | 94.50 | 92.50 | 57.70 | 74.05 |
| METEOR | 56.70 | 97.60 | 94.20 | 63.40 | 77.98 |
| CIDEr | 53.00 | 98.00 | 91.50 | 64.50 | 76.75 |
| SPICE | 52.60 | 93.90 | 83.60 | 48.10 | 69.55 |
| TIGEr* | 56.00 | 99.80 | 92.80 | 74.20 | 80.70 |
| CLIP-S* | 56.50 | 99.30 | 96.40 | 70.40 | 80.70 |
| RefCLIP-S* | 64.50 | 99.60 | 95.40 | 72.80 | 83.10 |
| CLAIR | | | | | |
| + GPT3.5 | 52.40 | 99.50 | 89.80 | 73.00 | 78.67 |
| + Claude | **57.90** | 98.50 | 91.30 | 62.90 | 77.65 |
| + PaLM | 54.70 | 98.30 | 87.30 | 64.00 | 76.08 |
| CLAIR$_E$ | 57.70 | **99.80** | **94.60** | **75.60** | **81.93** |

Table 4: Pearson correlation with human judgments when evaluating sets of captions on MS-COCO ($N = 794$).

| Measure | Coverage$_{p\text{-value}}$ | Correctness$_{p\text{-value}}$ |
|---|---|---|
| BLEU@4 | 0.004 $_{0.816}$ | 0.003 $_{0.888}$ |
| ROUGE-L | 0.011 $_{0.563}$ | 0.038 $_{0.184}$ |
| METEOR | 0.016 $_{0.398}$ | 0.006 $_{0.765}$ |
| CIDEr | 0.004 $_{0.844}$ | 0.026 $_{0.173}$ |
| TRM-METEOR | 0.128$_{<0.001}$ | 0.108$_{<0.001}$ |
| TRM-BLEU | 0.127$_{<0.001}$ | 0.151$_{<0.001}$ |
| MMD-BERT | 0.129$_{<0.001}$ | 0.124$_{<0.001}$ |
| FID-BERT | 0.081 $_{0.011}$ | 0.098$_{<0.001}$ |
| CLAIR | | |
| + GPT3.5 | **0.195** $_{0.011}$ | **0.187** $_{0.014}$ |
| + Claude | 0.110 $_{0.099}$ | 0.124 $_{0.145}$ |
| + PaLM | 0.129 $_{0.081}$ | 0.085 $_{0.172}$ |
| CLAIR$_E$ | 0.183 $_{0.027}$ | 0.156 $_{0.018}$ |
| Inter-Human | 0.225$_{<0.001}$ | 0.274$_{<0.001}$ |

**System-level human correlation:** In addition to computing the sample-level correlation on the MS-COCO dataset, we use the annotations from the five models considered by Rohrbach et al. (2018) to compute the system-level correlation. For each of the methods, we compute the mean human score on the test samples, and mean metric score on the test samples, followed by the Kendall's rank correlation coefficient (Kendall's tau, strength of ordinal association) between these values (the set of five mean human scores, and the set of five metric scores). Our results, given in Table 2, demonstrate that CLAIR ranks the five methods in a novel way that is more accordant with human rankings of the methods. These results further suggest that CLAIR has the potential to redefine which methods are preferable to humans compared to existing n-gram approaches.

**Decision Making:** In addition to evaluating the correlation with human judgments, we also evaluate the capability of the measure to perform discrimina-

tive analysis. The PASCAL-50S dataset (Vedantam et al., 2015) contains a set of 4000 human-annotated caption pairs. For each pair of captions, humans label which caption in the pair is closest to the reference set for the image. The caption pairs fall into four groups: "HC:" two human-written captions matching the image, "HI:" one human caption, and one machine-generated caption, with only one matching the image, "HM:" a matching human caption and a matching machine-generated caption and "MM:" two matching machine-generated captions. See Appendix B for more dataset information.

The performance on PASCAL-50S is given in Table 3. We can see that CLAIR$_E$ outperforms all existing text-only measures (e.g., by 5.18% overall score over CIDEr), and in many cases, even outperforms measures that have access to the image at test time. Note that it is relatively weaker than image-augmented models in the HC setting; however, since both captions are correct, the model often cannot judge which is better purely the text. Models such as RefCLIP-S that have access to the image are naturally better discriminators in this case. We suspect that CLAIR's discriminative performance could be further improved by giving the LLM a choice between the two captions; however, we leave this optimization to future work.

**Groups of Captions:** While CLAIR is capable of comparing a single candidate caption to a set of reference captions, it is also capable of comparing *sets* of candidate captions to sets of reference captions. This task is necessary when evaluating the ability of a model to generate captions that are diverse and that fully describe the conditional text

distribution. We evaluate on the COCO-Sets dataset (Chan et al., 2022), 794 caption sets rated by AMT workers on two scales: how closely a candidate set matches the reference set in terms of both correctness and content coverage (See Appendix B for details). The results of this experiment are given in Table 4. We can see that CLAIR outperforms well when measuring the quality of a group of captions, and approaches the inter-human correlation on the (very) challenging task. CLAIR also outperforms TRM-METEOR and TRM-BLEU (Chan et al., 2022), suggesting that LLMs can judge both the content and diversity of the caption sets.

## 4 Limitations

While CLAIR correlates well with human judgments of caption quality, it has several limitations:

**Non-Determinism and Parsing Errors:** Because CLAIR depends on the output of a language model, the measure can be non-deterministic and noisy. For instance, it may fail to elicit a judgment (e.g., "As an AI language model, I cannot see, and thus, cannot determine if the image captions match the references"), or rarely, generate malformed JSON output. To address these issues, we perform multiple queries to the LLM, sometimes at higher temperatures if necessary. As a consequence, the measure may differ between runs, although we found the variance to be relatively insignificant ($< 0.01$ in many of the experiments). Additionally, since the language models used are not open-source, the models are subject to arbitrary change, replacement, or removal, which limits the efficacy of the measure as a long-term comparable measurement. We hope that increasing open access to language models with efforts such as Koala (Geng et al., 2023) and Vicuna (Chiang et al., 2023), will help to alleviate these challenges in the future.

**Increased Cost:** CLAIR relies on language models which contain many billions of parameters. These language models have not only monetary cost but also human and environmental costs (Bender et al., 2021) which can reduce its utility as a target during training, such as for self-critical sequence training (Rennie et al., 2017). While API-based LLMs may be considered costly, even open-source LLMs have a cost (which can often be hard to quantify). CLAIR on the MS-COCO dataset uses an average of 226.148 tokens per sample (on OpenAI's API), representing a cost of $0.0067 per sample (GPT-4), or $0.00033 per sample (GPT 3.5). For PALM,

this drops to $0.000113 per sample. We hope that over time, advances in LLM inference (such as quantization and distillation), coupled with improvements in architecture will continue to yield lower-cost alternatives with strong performance on the caption evaluation task.

**Hallucination:** While CLAIR does suffer from potential hallucination, we strongly believe that this weakness does not diminish the fact that CLAIR still correlates strongly with human judgment. In CLAIR, hallucinations in the score manifest as "incorrect" judgements of similarity, while hallucinations in the explanations manifest as poorly grounded explanations of the score/quality. Hallucinations in the score should be considered false negatives (blind spots instead of hallucinations). In the case of hallucinations in the explanations, such hallucinations may lead to misinterpretation, but arguably less misinterpretation than a black box method, and may even indicate misunderstandings in the model. Hallucination is a well-known challenge of current LLMs and is the subject of a great amount of research on instruction-tuning, RLHF, RLAIF, and other methods. As hallucination and instruction-following performance of the base models improves, CLAIR inherit similar improvements.

**Explainability:** While CLAIR generates explanations for each rating, CLAIR has no strict scoring rubric. Much like human judgments, there is no direct way of attributing changes in score to changes in caption quality. For similar reasons, it is difficult to evaluate the quality of the generated explanations. Qualitatively, the explanations are generally reasonable and consider multiple axes of judgment.

## 5 Conclusion

This work introduces CLAIR, an LLM-based evaluation measure for image captioning. CLAIR's superior performance compared to highly-engineered measures indicates a remarkable fact: LLMs are well aligned with human judgments of caption quality, even more so than some measures designed specifically for semantic similarity. CLAIR is only a glimpse into how LLMs can be used for evaluation tasks, and image captioning is only the beginning. We hope that our work will inspire further exploration of similar measures in other vision and language domains, such as visual storytelling (Huang et al., 2016), where human evaluation of generated text remains a challenging task.

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

# Appendix

## A    Acknowledgements

We thank Suhong Moon and Kate Saenko for their helpful comments on the work. Authors, as part of their affiliation with UC Berkeley, were supported in part by the NSF, DoD, and/or the Berkeley Artificial Intelligence Research (BAIR) industrial alliance program, as well as gifts from Anyscale, Astronomer, Google, IBM, Intel, Lacework, Microsoft, Mohamed Bin Zayed University of Artificial Intelligence, Samsung SDS, Uber, and VMware.

## B    Additional Experimental Details

In this section, we provide several additional details for the experiments in section 3 run with the CLAIR measure.

### B.1    Input Prompt Formatting

The CLAIR prompt is given in its entirety in Figure 1. During run-time, candidate and reference captions are prefixed with a "- " and inserted into the prompt, one per line. The resulting query is passed to the large language model. In addition, for models which were not RLHF-tuned to perform conversation (such as PaLM), we found that it was helpful to append an additional prefix `{"score":` to the beginning of the output, to encourage the correct output formatting. CLAIR is surprisingly simple: it uses no in-context examples (is entirely zero-shot), and default inference parameters for the APIs. The model checkpoint metadata is generally unknown (as the APIs are somewhat fluid and evolving).

### B.2    LLM Output Post-Processing

Because CLAIR relies on an LLM to produce output, there is no guarantee that the output will be in the format we expect (i.e. valid, parsable JSON). To extract both the score and the reason, we first extract the first set of paired braces from the output of the LLM and attempt to parse the result as JSON.

In most cases ($99.997\%$ for GPT-3, $99.991\%$ for Claude, and $99.94\%$ for PaLM during the course of our experiments), this is successful, and the score and reason are returned. In the case that the JSON output is malformed, we attempt to extract any sequence of digits from the LLM to use as a score, and set the reason to "Unknown." When this fails, as can be the case when the models produce an output such as "As an AI language model, I cannot see, and thus, cannot determine if the image captions match the references", we retry the prompt at a higher temperature ($t = 1.0$) several times. Failing this (which occurred only three times in the entire evaluation of this paper, across several hundred thousand calls), we set the score to 0 for the caption.

### B.3    Datasets

In this section, we provide additional detail regarding the datasets used in the evaluations in section 3.

**COMPOSITE:** The COMPOSITE dataset (Aditya et al., 2015) contains machine-generated test captions for 3995 images spread across the MS-COCO (Xu et al., 2016), Flickr8K (Mao et al., 2014) and Flickr30k (Young et al., 2014) datasets. Each image has three test captions, one written by a human, and two that are model generated. The candidate captions are graded by annotators on Amazon Mechanical Turk (AMT) on a scale of 1 (not relevant) to 5 (very relevant). Inter-human correlations are not available for this dataset.

**Flickr8K-Expert:** The Flickr8K-Expert dataset (Hodosh et al., 2013) contains 5822 captions associated with 1000 images. The dataset is annotated with expert human judgments of quality, where images are rated from 1 (caption is unrelated to the image) to 4 (caption describes the image without errors). Unlike the composite and MS-COCO datasets, the captions here are selected using an image retrieval system, instead of generated using a learned image captioning model. Following Jiang et al. (2019), we exclude any candidate captions that overlap the reference set.

**MS-COCO:** Following experiments by Rohrbach et al. (2018), we compute the sample-level correlation between our method and human ratings on a 500-image subset of the MS-COCO Karpathy test set. Each image in the subset contains candidate captions generated by 5 models, and each caption is labeled with the average three human

ratings generated by AMT workers which range from 1 (very bad) to 5 (very good). Inter-human correlations are not available for this dataset.

**PASCAL-50S:** PASCAL-50S contains 1000 images drawn from the PASCAL sentence dataset. Each image is associated with at least 50 (and as many as 120) reference captions. In addition to the reference captions, PASCAL-50S contains a set of 4000 human annotated image/caption pairs containing an image, and two candidate captions. The caption pairs fall into four groups:

1. HC: In the HC group, both captions in the pair are human written, and describe the content of the target image correctly.

2. HI: In the HI group, both captions in the pair are human written, but one caption correctly describes the content of the image, and the other caption describes the content of a different image.

3. HM: In the HM group, one caption is written by a human, and one caption is written by a machine, but both correctly describe the content of the image.

4. MM: In the MM group, both captions are written by a machine, and both correctly describe the image content.

In PASCAL-50S, the task is to decide which caption in the pair humans prefer more (a subjective task, hopefully indicating caption quality). Following previous work (Jiang et al., 2019; Hessel et al., 2021), we limit the number of reference sentences to five during evaluation.

**COCO-Sets:** The COCO-Sets dataset (Chan et al., 2022) is a set of samples that are designed to evaluate the correlation of distribution-aware image captioning measures with human judgments of distributional distance. In this dataset, humans were presented with two candidate caption sets (two image captioning models, OFA (Wang et al., 2022) and BLIP (Li et al., 2022) using different temperatures), and asked which candidate caption set correlated better with a reference caption set on two measures: how much they overlapped factually (correctness), and how much information they provided about the references (coverage). It consists of 794 AMT worker-generated judgments of caption quality for images in the MS-COCO dataset.