# OpenReview forum: "CLAIR: Evaluating Image Captions with Large Language Models"
_EMNLP/2023/Conference — EMNLP 2023 Main_

### Official Review · Reviewer_Cmgu · 2023-07-30

**Typos Grammar Style And Presentation Improvements:** The gold reference could be included …
**Soundness:** 3

**Excitement:**

3: Ambivalent: It has merits (e.g., it reports state-of-the-art results, the idea is nice), but there are key weaknesses (e.g., it describes incremental work), and it can significantly benefit from another round of revision. However, I won't object to accepting it if my co-reviewers champion it.

**Missing References:**

[1] Towards Understanding Sample Variance in Visually Grounded Language Generation: Evaluations and Observations, EMNLP 2020.

[2] Learning to Rank Visual Stories from Human Ranking Data, ACL 2022.

[3] Visual storytelling, NAACL 2016.

[4] No Metrics Are Perfect: Adversarial Reward Learning for Visual Storytelling, ACL 2018.

[5] Knowledge-enriched visual storytelling, AAAI 2020.

**Paper Topic And Main Contributions:**

This paper introduces a novel usage of LLMs, wherein they are employed to evaluate model-generated image captions. Extensive experiments with strong baselines demonstrate the superiority of this method, and the results indicate that their proposed method, CLAIR, strongly aligns with human judgment.


**Questions For The Authors:**

(1) As CLAIR scores generations solely based on texts, I am wondering what makes it unique and applicable to image captioning. Could it be extended to other text-based generation tasks? The paper should strengthen the motivation of using CLAIR specifically for the image captioning task.

(2) Is it applicable to other image-to-text generation tasks? Could it score well with image-to-text generations with long text targets, such as visual storytelling [3]? Judging long texts is still a challenging problem.

(3) As the authors mentioned in the introduction section, each evaluation metric measures different aspects of captions. Have the authors tried using different prompts that align with the same aspects as other metrics to enable a fairer comparison?

(4) Have the authors considered pair-wise ranking? Some papers [4,5] have shown that pair-wise ranking is more stable than ranking through many candidates at once.

(5) Have the authors experimented with other instructions (e.g., as shown in Figure 1) to observe the differences?

**Reasons To Accept:**

(1) The paper is very well-written and easy to follow. The organization of the manuscript effectively follows their contributions.

(2) This paper presents interesting aspects of using LLMs to assess the quality of generations.

(3) The authors conduct extensive experiments to demonstrate the improvements of this new evaluation method.

(4) The method not only provides a score for the image caption but also offers a reason for why this judgment is made.

(5)T hough the metric does not consider images as input, it is interesting to see that the results outperformed other metrics that do consider images

**Reasons To Reject:**

(1) Though the paper introduces an interesting usage of LLM and demonstrates its superiority, the motivation for considering the image captioning task should be strengthened since the judgment of their method is solely based on texts.

(2) Developing captioning models based on this metric could be a bit costly for users.

(3) The paper should consider mentioning other multimodal generation tasks and evaluation tasks [1,2] (see missing references).

**Reproducibility:**

2: Would be hard pressed to reproduce the results. The contribution depends on data that are simply not available outside the author's institution or consortium; not enough details are provided.

**Reviewer Confidence:**

4: Quite sure. I tried to check the important points carefully. It's unlikely, though conceivable, that I missed something that should affect my ratings.

---

> ### Author Rebuttal · Authors · 2023-08-29
>
> We appreciate the reviewer’s comments, and that the reviewer highlights our extensive experiments, and easy to understand methods for evaluation of the image captioning methods. We'd like to take this opportunity to respond to some of the weaknesses and questions discussed by the reviewer.
>
> **Motivation for Image Captioning/Could CLAIR be extended to other tasks:** While we do believe that CLAIR is applicable to domains outside of image captioning, in this short paper we decided to focus on image captioning as it is well within the scope of the authors’ expertise, as well as clearly indicates the benefits of a flexible, LLM-enabled evaluation method. Further, automatically evaluating the quality of image captions is a sufficiently challenging problem, and a good forum for understanding the potential of such LLM-enabled measures. There are many dimensions to consider, such as grammatical quality, semantic relevance, correctness, and specificity, which all lead to a good forum to showcase the potential approach.
>
> We strongly believe that introducing CLAIR as an image captioning measure does not preclude the application of CLAIR to other tasks such as visual storytelling (which we will discuss as a potential application, as well as mention related references for, in the camera ready paper), and instead, we hope that introducing CLAIR inspires the use of LLM-enabled measures across a wide range of potential applications!
>
> **Other Prompts for CLAIR/Directed Alignment to other measures:** In this short paper, we primarily focused on CLAIR as a replacement for/augmentation to existing n-gram measures for semantic similarity. While we did experiment somewhat in terms of finding an optimal prompt for this task, we did not have any significant interpretable findings regarding the choice of prompt. Our experiments show that with the existing prompt (Figure 1), we are already able to outperform general measures in terms of overall human correlation, which we do believe is a fair approach when comparing the metrics as general purpose measures.
>
> We do believe that flexibly adapting the prompt can serve as an interesting tool to use for targeted evaluation, for example, we could ask only about certain facets of the image, such as the background of the image, or only ask the model to measure the semantic distance between nouns (ignoring adjectives). We strongly believe that it is interesting and exciting future work to explore “directed” versions of the CLAIR measure, however such work is outside the scope of the current introductory work.
>
> **Cost of LLMs:** While API-based LLMs may be considered costly, even open-source LLMs have a cost (which can often be hard to quantify). CLAIR on the MS-COCO dataset uses an average of 226.148 tokens per sample (on OpenAI’s API), representing a cost of \\$0.0067 per sample (GPT-4), or \\$0.00033 per sample (GPT 3.5). For PALM, this drops to \\$0.000113 per sample. We hope that over time, advances in LLM inference (such as quantization and distillation), coupled with improvements in architecture will continue to yield lower-cost alternatives with strong performance on the caption evaluation task. Such a trend is already evident: models such as Llama-2 (released after the submission of this short paper) demonstrate performance on par with Chat-GPT, while remaining free and accessible. Thus, while currently LLMs are a costly tool, we do not believe that long term LLM cost is a fundamental weakness of CLAIR. We will make this clear in the limitations section of the paper, where we already discuss the weaknesses of relying on LLM approaches.
> Pairwise Ranking: In this short paper we did not consider a pairwise ranking approach to stabilize the performance of the metrics (either CLAIR or other measures). We believe that this is an excellent direction for further research: like humans, do LLMs also benefit from the reduction in context/specificity provided by pairwise ranking.
>
> **Other Comments:** We will include the gold references for the captions In Figure 2 in the camera ready.

---

### Official Review · Reviewer_2NfD · 2023-08-05

**Soundness:** 3

**Excitement:**

3: Ambivalent: It has merits (e.g., it reports state-of-the-art results, the idea is nice), but there are key weaknesses (e.g., it describes incremental work), and it can significantly benefit from another round of revision. However, I won't object to accepting it if my co-reviewers champion it.

**Paper Topic And Main Contributions:**

This paper proposes an approach for evaluating image captioning methods by taking advantage of a large language model.  Specifically, they formulate the evaluation protocol for image captioning as a question that they pose to the LLM to answer.  The authors demonstrate that their approach provides more human-aligned scores than metrics proposed by prior work.

**Reasons To Accept:**

This paper has demonstrated that their approach provides more human aligned scores than on several popular benchmarks

The set of related tasks to image captioning is large and this work could potentially provide a good metric for many of them

The paper is relatively easy to follow and understand

**Reasons To Reject:**

1.  It isn't clear that the proposed metric will provide a more accurate ranking of methods.  In particular, I would wonder if the ranking of models produced by the proposed metric is different than the metrics we already use for image captioning

2. The authors use relatively easy datasets for evaluating their model. However, if they demonstrated their approach could also be used to evaluate models on datasets like Winoground or BISON, where there are meant to be similar images with captions that have a lot of similar words, then the metric would be far more beneficial for future work on an area where image captioning methods are known to work poorly.

3. The authors note that some LLMS do not provide good scores, but to not provide any insight that would enable us to understand the reason for this discrepancy

4. It isn't clear that the proposed metric would be useful for clearly wrong captions given an image.  For example, if one were to replace "dog" for "cat" in a caption, would the metric notice? This would be somewhat addressed by the second weakness, but it isn't clear even in settings that the current metrics would be able to distinguish between models that the proposed one would.

**Reproducibility:**

4: Could mostly reproduce the results, but there may be some variation because of sample variance or minor variations in their interpretation of the protocol or method.

**Reviewer Confidence:**

4: Quite sure. I tried to check the important points carefully. It's unlikely, though conceivable, that I missed something that should affect my ratings.

---

> ### Author Rebuttal · Authors · 2023-08-29
>
> We appreciate the reviewer’s comments, and that the reviewer highlights our human-aligned, transferable, and easy to understand methods for evaluation of the image captioning task. We'd like to take this opportunity to respond to some of the weaknesses discussed by the reviewer.
>
> **Ranking of methods:** While we do not explicitly evaluate the ranking of the method on all of our datasets, in Table 2/Section 3 we provide system-level correlations with human judgment on the MS-COCO measure. The system-level correlation results, which indicate the correlation between the methods’ rankings of different models and human rankings of the same models, show that CLAIR significantly outperformed the baseline measures with respect to human judgements of model quality. In general, we also find that sample-level correlation is well correlated to human judgment (Table 1), which we strongly believe is evidence that CLAIR is capable of ranking methods, in addition to sample captions.
>
> **Easy Datasets:**  We agree that the datasets we evaluate on are not specialized to fine-grained image differentiation. As the reviewer mentions, both Winoground and BISON are designed for evaluating a model’s ability to distinguish fine-grained image concepts (they consist of one or two images, and a challenging single caption to match to the image). CLAIR, however, is intended as a replacement for reference-based measures of caption quality, which means that evaluating CLAIR on these datasets would measure the distance between the ground truth COCO captions, and the candidate captions in Winoground/BISON. In a way, this measures the performance of the “ground truth” model, which is likely somewhat divergent from the quality of the metric, as it is generally understood that even reference image captions do not cover the full range of content present in the images [4,5,6].
>
> **Clearly wrong captions:** We understand the concern that a soft measure such as CLAIR will fail to recognize clearly incorrect captions (as this is often a weakness of measures based on embeddings, such as BERT-Score or CLIPScore). To evaluate the quality of CLAIR on such replacements, we evaluated the accuracy of CLAIR when applied to the FOIL dataset of hallucinations. FOIL is a dataset where candidate captions are constructed by replacing single nouns in correct captions with semantically close mismatches. We found that CLAIR (on average) detected 93.6% of the semantic mismatches, which is very close to human performance of 94.52%, and above RefCLIPScore [3] (92.6%) and even CHAIRs [2] (92.5%, a metric designed explicitly for hallucination detection in captions)
>
> **LLM Scores:** In general, we found that the quality of the CLAIR scores produced by the LLM is closely related to the language model’s ability to perform reasoning tasks, such as question answering, as well as the model’s ability to respond to questions in the requested format. We strongly believe that it is largely a coincidence that this set of models overlaps with the set of API-based models (compared to free models such as Llama/Vicuna). A detailed investigation of why CLAIR works is certainly interesting: what parts of the training datasets correlate strongly with CLAIR performance, and why is it that CLAIR is able to approximate human judgements of semantic correlation so well? These are challenging questions to answer without significant control of the underlying language model, and while fundamental, we believe they are outside the scope of this short paper, which takes the first step towards quantifying and demonstrating the performance of LLMs in a zero-shot manner on semantic judgment tasks for image captioning applications.
>
> [1] Shekhar, Ravi, et al. "Foil it! find one mismatch between image and language caption." arXiv preprint arXiv:1705.01359 (2017).
> [2] Rohrbach, Anna, et al. "Object hallucination in image captioning." arXiv preprint arXiv:1809.02156 (2018).
> [3] Hessel, Jack, et al. "Clipscore: A reference-free evaluation metric for image captioning." arXiv preprint arXiv:2104.08718 (2021).
> [4] Caglayan, Ozan, Pranava Madhyastha, and Lucia Specia. "Curious case of language generation evaluation metrics: A cautionary tale." arXiv preprint arXiv:2010.13588 (2020).
> [5] Yeh, Yi-Ting, Maxine Eskenazi, and Shikib Mehri. "A comprehensive assessment of dialog evaluation metrics." arXiv preprint arXiv:2106.03706 (2021).
> [6] Chan, David M., et al. "Distribution aware metrics for conditional natural language generation." arXiv preprint arXiv:2209.07518 (2022).

---

### Official Review · Reviewer_kf2G · 2023-08-10

**Soundness:** 4

**Excitement:**

3: Ambivalent: It has merits (e.g., it reports state-of-the-art results, the idea is nice), but there are key weaknesses (e.g., it describes incremental work), and it can significantly benefit from another round of revision. However, I won't object to accepting it if my co-reviewers champion it.

**Paper Topic And Main Contributions:**

The paper introduces CLAIR, a new image captioning evaluation metric.

CLAIR uses  frozen LLMs to evaluate the generated caption by comparing with reference captions. Specifically, they give a prompt to LLMs includes 1) candidate captions, 2) reference captions, and 3) task description, where they ask LLM to predict a score within 1-100 range along with text reasoning for the scoring.

They experiment with 3 LLMs: GPT-3.5 / Claude / PaLM and compare with existing metrics such as text-only metrics (e.g., BLEU/ROUGE/BertScore) and image-text metrics (e.g., CLIPScore).

They provide quantitative analysis including sample-level correlation (Composite/Flickr8k/COCO), system level correlation (COCO), decision making (Pascal-50S), and group of captions (COCO-sets). Overall, the proposed CLAIR metric achieves higher correlation metrics then the baseline metrics.

**Questions For The Authors:**

- Q1: Can you clarify Pascal-50S dataset description (L189-204)? I checked the appendix A L537-557, but I still can’t fully understand how the 4 groups are constructed. Showing some examples and high-level explanation of the characteristics of 4 groups would be help readers.

**Reasons To Accept:**

- S1: simple but effective implementation.

- S2: strong empirical demonstration of higher human correlation than the existing metrics.

- S3: providing quantitative experiments results on multiple different evaluation benchmarks.

**Reasons To Reject:**

- W1: weak technical novelty. There are already captioning evaluation metrics using LM (e.g., BertScore).

- W2: weak baselines. The sizes of the backbone models used in the baseline model-based evaluation (e.g., BertScore, CLIPScore) are much smaller than the LLMs used in CLAIR. As the authors used large LMs for CLAIR, we can also compare with larger components for the baseline metrics like CLIPScore. Also recently proposed text-to-image evaluation metrics that are using LLMs such as TIFA/VPEval/SeeTRUE can be directly used as baselines.

- W3: missing ablation studies: The authors do not provide any studies other than human correlations. Different design choices (e.g., different in-context examples, different prompt structure, use of reasoning, different reasoning) can affect different results. It would be also interesting how well smaller open-source language models (e.g., llama1/2, flan-t5, vicuna) can perform in CLAIR.

- W4: API cost. As mentioned in L277-289, evaluation of these models involves payment to their LLM host companies. In addition to the RL use cases mention by authors (L284-286), usually image captioning pipeline developments involve a lot of evaluation runs (e.g., evaluating on a validation set for each epoch). It would be hard for many academic researchers to repeatedly use CLAIR during model development.

- W5: missing details: It would be hard to replicate the authors experiments given the information in the paper. Missing details include: number of in-context examples, LLM inference hyperparameters, model checkpoint metadata.

**Reproducibility:**

4: Could mostly reproduce the results, but there may be some variation because of sample variance or minor variations in their interpretation of the protocol or method.

**Reviewer Confidence:**

4: Quite sure. I tried to check the important points carefully. It's unlikely, though conceivable, that I missed something that should affect my ratings.

---

> ### Author Rebuttal · Authors · 2023-08-29
>
> We appreciate the reviewer’s comments, and that the reviewer highlights our extensive experiments and simple and effective method for evaluation of the image captioning task. We'd like to take this opportunity to respond to some of the weaknesses and questions discussed by the reviewer.
>
> **Clarifying Pascal 50S Experiments:** We apologize for any confusion with the PASCAL-50S dataset, and will seek to clarify the dataset in the camera ready version of the paper. PASCAL-50S is a dataset consisting of images paired with at least 50 reference captions. In addition to the reference captions, there are 4000 human annotated image/caption pairs containing an image, and two candidate captions. There are four types of caption pair:
>
> - HC: In the HC group, both captions in the pair are human written, and describe the content of the target image correctly.
> - HI: In the HI group, both captions in the pair are human written, but one caption correctly describes the content of the image, and the other caption describes the content of a different image.
> - HM: In the HM group, one caption is written by a human, and one caption is written by a machine, but both correctly describe the content of the image.
> - MM: In the MM group, both captions are written by a machine, and both correctly describe the image content.
>
> In PASCAL-50S, the task is to decide which caption in the pair humans prefer more (a subjective task, hopefully indicating caption quality). We will add examples, as well as seek to clarify this in the camera ready version of the paper. We are also happy to answer any further questions during the discussion period.
>
> **API Cost:** While API-based LLMs may be considered costly, even open-source LLMs have a cost (which can often be hard to quantify). CLAIR on the MS-COCO dataset uses an average of 226.148 tokens per sample (on OpenAI’s API), representing a cost of \\$0.0067 per sample (GPT-4), or \\$0.00033 per sample (GPT 3.5). For PALM, this drops to \\$0.000113 per sample. We hope that over time, advances in LLM inference (such as quantization and distillation), coupled with improvements in architecture will continue to yield lower-cost alternatives with strong performance on the caption evaluation task. Such a trend is already evident: models such as Llama-2 (released after the submission of this short paper) demonstrate performance on par with Chat-GPT, while remaining free and accessible. Thus, while currently LLMs are a costly tool, we do not believe that long term LLM cost is a fundamental weakness of CLAIR. We will make this clear in the limitations section of the paper, where we already discuss the weaknesses of relying on LLM approaches.
>
> **Missing Ablations:** While we do not provide any additional experiments besides our extensive human correlation experiments, we believe that human experimentation remains the gold standard for evaluating semantic similarity measures. We agree that exploring additional axes of the problem can lead to interesting effects: for example, we do not use any in-context learning (i.e. no examples), however biasing the model with additional context could lead to different results. While we do not ablate the choice of prompt structure, the proposed structure (Figure 1) demonstrates that even with simple requests, we are able to achieve strong correlation – we don’t believe that an ablation of the prompt structure would lead to materially interesting results, as there would be no way to scientifically demonstrate models are using particular parts of the prompt effectively. We agree that additional experimentation with open-source models would be an interesting augmentation, and we will aim to include as many as possible in the camera ready version of the paper, however we strongly believe that the primary contribution of CLAIR lies outside the specific LLM implementation.
>
> **Weak Baselines:** While the LLMs that we leverage for CLAIR are larger than those used for BERT-Score/CLIPScore, we don’t believe that developing a entirely new metric leveraging a larger LLM in the style of BERTScore/CLIPScore is within the scope of this short paper. Furthermore, CLAIR represents a novel way of using LLMs, i.e. just ask the LLM to produce a token-level score, as opposed to using an implicit embedding in the LLM’s latent space, which we believe sets the approach apart from existing LLM-based methods.
>
> We do agree that some recent reference-free measures (published in May of this year, less than three months before the submission deadline) such as TIFA, VPEval, and SeeTRUE are interesting potential comparisons. These reference-free measures serve as interesting potential concurrent work, and we will add them to the related work section, and aim to generate comparisons and human evaluations of the method if possible. We don’t believe, however, that the existence of additional reference-free measures diminish the fact that CLAIR represents a surprisingly simple and effective measure for measuring semantic similarity in a reference-based setting (none of TIFA, VPEval, or SeeTrue leverage reference captions), and as we discuss in the paper, we strongly believe that there is a place for both reference-based and reference-free evaluation measures.
>
> **Weak Technical Novelty:** While CLAIR is primarily a prompting paper, we believe that this short paper is of significant interest to the community. CLAIR is the first paper which demonstrates (with extensive human studies and experimentation) that LLMs correlate stronger with human judgment than any existing reference-based measure when evaluating image captioning tasks. Furthermore, CLAIR is one of the first papers to demonstrate that LLMs are capable of producing a token-based score of captions, meaning that methods such as CLAIR are usable even when underlying embeddings are unobtainable (such as is often the case with proprietary algorithms/models). Finally, CLAIR is, to our knowledge, the only text similarity measure which is capable of producing an interrogative explanation of its score, a hugely important tool which can be used for diagnostic and analysis purposes.
>
> **Missing Details:** CLAIR is surprisingly simple: it uses no in-context examples (is entirely zero-shot), and default inference parameters for the APIs. The model checkpoint metadata is generally unknown (as the APIs are somewhat fluid and evolving). We will clarify this in the main paper.
>
> [1] Hessel, Jack, et al. "Clipscore: A reference-free evaluation metric for image captioning." arXiv preprint arXiv:2104.08718 (2021).

---

### Official Review · Reviewer_kPSe · 2023-08-11

**Soundness:** 3

**Excitement:**

3: Ambivalent: It has merits (e.g., it reports state-of-the-art results, the idea is nice), but there are key weaknesses (e.g., it describes incremental work), and it can significantly benefit from another round of revision. However, I won't object to accepting it if my co-reviewers champion it.

**Paper Topic And Main Contributions:**

This paper proposes CLAIR, an LLM based novel evaluation measure for image captioning. The authors found that CLAIR demonstrates a stringer correlation with human judgements of caption quality compared to existing measures. It also shows that how language models can evaluate vision-language tasks. The LLM's can provide not only scalar ratings but also corresponding reasoning for a given rating, offering a valuable combination of accuracy & interpretability.



**Questions For The Authors:**

Q1. How will you tackle hallucinations in CLAIR as it uses LLMs.

**Reasons To Accept:**

The following are the reasons to accept the paper --

1. The paper is well written and easy to follow.

2. The authors proposes a novel approach for evaluation using LLM for image-captioning task.

3. CLAIR achieves relative correlation improvement over SPICE of 39.6% on Flickr8k dataset and over image-augmented methods such as RefCLIP of 18.3%


**Reasons To Reject:**

The following are the weakness of the paper -

1. LLM's are costly to run and evaluate and are constrained by resources. So, in real scenarios it might be difficult to use this approach.

2. Evaluation of generated explanations by LLM's are difficult to evaluate.

3. The evaluation might suffer from hallucinations produced by LLM's.

**Reproducibility:**

1: Could not reproduce the results here no matter how hard they tried.

**Reviewer Confidence:**

3: Pretty sure, but there's a chance I missed something. Although I have a good feel for this area in general, I did not carefully check the paper's details, e.g., the math, experimental design, or novelty.

---

> ### Author Rebuttal · Authors · 2023-08-29
>
> We appreciate the reviewer’s comments, and that the reviewer highlights our novel, and powerful methods for evaluation of the image captioning task. We'd like to take this opportunity to respond to some of the weaknesses discussed by the reviewer.
>
> **Hallucinations in CLAIR:** While CLAIR does suffer from potential hallucination, we strongly believe that this weakness does not diminish the fact that CLAIR still correlates strongly with human judgment (even more so than traditional reference-based natural language evaluation measures).  In CLAIR, hallucinations in the score manifest as “incorrect” judgements of similarity, while hallucinations in the explanations manifest as poorly grounded explanations of the score/quality. Hallucinations in the score should be considered false negatives (blind spots instead of hallucinations). In the case of hallucinations in the explanations, such hallucinations may lead to misinterpretation, but arguably less misinterpretation than a black box method, and may even indicate misunderstandings in the model. Further, hallucination is a well-known challenge of current LLMs and is the subject of a great amount of research on instruction-tuning, RLHF, and other methods. As hallucination and instruction-following performance of the base models improves, we should inherit similar improvements in CLAIR. To further address this concern, we will add additional discussion about hallucination, and indicate some of the steps that can be taken to reduce hallucination.
>
> **LLMs are Costly:** While API-based LLMs may be considered costly, even open-source LLMs have a cost (which can often be hard to quantify). CLAIR on the MS-COCO dataset uses an average of 226.148 tokens per sample (on OpenAI’s API), representing a cost of \\$0.0067 per sample (GPT-4), or \\$0.00033 per sample (GPT 3.5). For PALM, this drops to \\$0.000113 per sample. We hope that over time, advances in LLM inference (such as quantization and distillation), coupled with improvements in architecture will continue to yield lower-cost alternatives with strong performance on the caption evaluation task. Such a trend is already evident: models such as Llama-2 (released after the submission of this short paper) demonstrate performance on par with Chat-GPT, while remaining free and accessible. Thus, while currently LLMs are a costly tool, we do not believe that long term LLM cost is a fundamental weakness of CLAIR. We will make this clear in the limitations section of the paper, where we already discuss the weaknesses of relying on LLM approaches.
>
> **Generated Explanations are Hard to Evaluate:** We agree that evaluating the explanations generated by CLAIR can be challenging. CLAIR’s rate of hallucination is largely tied to the performance of the underlying LLM As LLMs evolve, we hope that the incidence of hallucination will be dramatically reduced. While one potential avenue for evaluation is human studies of the quality of the explanations, it’s not immediately obvious what constitutes a hallucination in a generated explanation. For example, if we define hallucination as “a description of something that is not grounded in the inputs”, a hallucination in the explanation could be an invaluable indicator of the uncertainty or trustworthiness of the actual score, and raise red flags that may otherwise go unnoticed if a reviewer was just presented with a black box number. Understanding how CLAIR hallucinates, and performing a deep dive into the ways that LLM hallucinations can indicate confidence and uncertainty is an important direction for future research. We will make this clear in the limitations and future work sections of the paper.

---

### Meta-Review · Area_Chair_xd5m · 2023-09-17

**Recommendation:** 4

**Metareview:**

Evaluating machine-generated image captions against reference captions has proven to be a difficult task. This paper proposes simply asking LLMs to score machine-generated captions versus reference captions, and shows that LLM performance aligns closely with human judgments.

Pros:
- Focused problem statement, appropriate scope for a short paper.
- Much better alignment with human judgment than previous metrics. Evaluating image captioning is an important task, and this work brings a methodology that outperforms existing metrics by a significant margin when it comes to aligning with human judgement.
- These findings are also relevant for other text evaluation challenges; the approach here is could be generalized beyond image captioning in future work (to be clear this is not necessary for this short paper). This is exciting.
- Well-written, easy to follow.

Cons:
- Using an LLM in the loop can be costly, so this metric may not be something that can practically be applied at training time. The authors  argue, sufficiently convincingly, that this will change over time and indeed is already changing with recent open-source models that have become available after the EMNLP deadline (and ACL's contemporaneous work policy).
- The authors also propose having the LLM explain its scores, but it's unclear if these explanations are trustworthy. This is ultimately somewhat irrelevant in the sense that the alignment with human judgment is what matters, however, as long as the explanations are not relied upon for anything.
- Depending on whether the LLM used to score is local or black-box+API-based, reproducibility of this metric may be concerning; for example, the model underlying an API call to a black-box model may change, even during scoring for a large set of captions, which could wreak havoc upon the results. This concern can be mitigated by using a stable model for every run.

---

### Decision · Program_Chairs · 2023-10-07

**Decision:**

Accept-Main

**Comment:**

Evaluating machine-generated image captions against reference captions has proven to be a difficult task. This paper proposes simply asking LLMs to score machine-generated captions versus reference captions, and shows that LLM performance aligns closely with human judgments.

Pros:
- Focused problem statement, appropriate scope for a short paper.
- Much better alignment with human judgment than previous metrics. Evaluating image captioning is an important task, and this work brings a methodology that outperforms existing metrics by a significant margin when it comes to aligning with human judgement.
- These findings are also relevant for other text evaluation challenges; the approach here is could be generalized beyond image captioning in future work (to be clear this is not necessary for this short paper). This is exciting.
- Well-written, easy to follow.

Cons:
- Using an LLM in the loop can be costly, so this metric may not be something that can practically be applied at training time. The authors  argue, sufficiently convincingly, that this will change over time and indeed is already changing with recent open-source models that have become available after the EMNLP deadline (and ACL's contemporaneous work policy).
- The authors also propose having the LLM explain its scores, but it's unclear if these explanations are trustworthy. This is ultimately somewhat irrelevant in the sense that the alignment with human judgment is what matters, however, as long as the explanations are not relied upon for anything.
- Depending on whether the LLM used to score is local or black-box+API-based, reproducibility of this metric may be concerning; for example, the model underlying an API call to a black-box model may change, even during scoring for a large set of captions, which could wreak havoc upon the results. This concern can be mitigated by using a stable model for every run.